# Wi-Fi Handshake: analysis of password patterns in Wi-Fi networks



Adrian Carballal[1], J. Pablo Galego-Carro[2], Nereida Rodriguez-Fernandez[3] and Carlos Fernandez-Lozano[1]

[1] Department of Computer Science and Information Technologies, Faculty of Computer Science, CITIC-Research Center of Information and Communication Technologies, Universidade da Coruña, A Coruña, A Coruña, Spain
[2] Computer Architecture Group, Faculty of Computer Science, Universidade da Coruña, A Coruña, Spain
[3] Department of Computer Science and Information Technologies, Faculty of Communication Science, CITIC-Research Center of Information and Communication Technologies, Universidade da Coruña, A Coruña, Spain

## ABSTRACT

This article seeks to provide a snapshot of the security of Wi-Fi access points in the metropolitan area of A Coruña. First, we discuss the options for obtaining a tool that allows the collection and storage of auditable information from Wi-Fi networks, from location to signal strength, security protocol or the list of connected clients. Subsequently, an analysis is carried out aimed at identifying password patterns in Wi-Fi networks with WEP, WPA and WPA2 security protocols. For this purpose, a password recovery tool called Hashcat was used to execute dictionary or brute force attacks, among others, with various word collections. The coverage of the access points in which passwords were decrypted is displayed on a heat map that represents various levels of signal quality depending on the signal strength. From the handshakes obtained, and by means of brute force, we will try to crack as many passwords as possible in order to create a targeted and contextualized dictionary both by geographical location and by the nature of the owner of the access point. Finally, we will propose a contextualized grammar that minimizes the size of the dictionary with respect to the most used ones and unifies the decryption capacity of the combination of all of them.

## INTRODUCTION

Although research on system security is necessary, it becomes indispensable when that system is "everywhere". This article explains why Wi-Fi is a ubiquitous technology today and how different iterations in password security protocols have made users the weakest link in the chain.

According to the Spanish National Statistics Institute (INE) (*INE, 2018*), as of 2018, 86.1% of Spanish households (more than 14 million) had broadband Internet access through the different technologies available: optical fiber or ADSL, mobile Internet (3G, 4G or 5G) or satellite. With the exception of mobile Internet, the technologies mentioned above are concerned with providing connection, not Internet access, for devices, so it is necessary to add an access point that communicates with them. Currently, the most widely

Corresponding author
Adrian Carballal,
adrian.carballal@udc.es

used device for this purpose is the router, which combines modem and router functions to create a private network to which various devices are connected, either *via* Ethernet cables or Wi-Fi.

The first routers incorporating Wi-Fi technology appeared in 2000, and it has since become the most popular way to connect to the Internet in the home and business environment. This popularity is clearly marked by two milestones: the term Wi-Fi was added to the Merriam-Webster dictionary in 2005 and by 2012, it had been implemented in 25% of households worldwide (*Alliance, 2019*).

Wi-Fi technology made it easy for a wide variety of devices to connect to the Internet without the need to be physically wired to the router. The advantages are clear: the cheaper and easier implementation of local area networks (LANs), proliferation of applications and mobile devices, possibility of creating spaces with immediate connectivity, and user mobility (*Castro, 2005*). However, connecting became riskier because physical access to the cable was no longer necessary in orther to attack a network, and the data were transmitted publicly, through the "air" (or, technically, the radio spectrum). Anyone within the range of a wireless network could try to access it.

The first security protocol that attempted to address this vulnerability in Wi-Fi networks was Wired Equivalent Privacy (WEP), launched at the same time as the first products certified by the Wi-Fi Alliance. However, by the end of 2001, the WEP protocol was already considered insecure (*Fluhrer, Mantin & Shamir, 2001*) and Wi-Fi Protected Access (WPA) was developed with the intention that any device supporting WEP encryption could support the new protocol with a simple firmware update, a compromise solution while the WPA2 standard was being developed. The WPA protocol, maintained certain problems and vulnerabilities of WEP (*e.g.*, chop chop attack (*Beck & Tews, 2008*) and packet injection (*Huang et al., 2005*)), but represented important advancements, along with the introduction of the four-way *handshake*.

In 2004, the first WPA2-compliant devices began to be certified, and starting in 2006, all new devices without Wi-Fi certification were required to implement it. In 2019, WPA2 became the best-known protocol, but it was neither the newest nor safest. During the last few years, vulnerabilities have been discovered (*e.g.*, the decryption of group keys (*Vanhoef & Piessens, 2016*) or the famous KRAK (*Vanhoef & Piessens, 2017*)) which, as with WPA, endanger the traffic between the client and access point, although they do not compromise the security of the key.

In January 2018, WPA version 3 was announced, with the intention of solving the problems discovered in WPA2. It is currently being implemented as an option by manufacturers. In this new version, the four-way *handshake* has been replaced by a protocol called "simultaneous peer-to-peer authentication" which promises to be resistant to *offline* dictionary attacks. In *offline* attacks, once the file is obtained containing the hash that is to be decrypted, there is no restriction regarding the number of possibilities that can be tested until it is achieved, contrary to what usually happens in Internet services. However, a vulnerability of this protocol has already been found in the WPA2 compatibility mode (*Vanhoef & Ronen, 2019*), which is that it would still allow *offline* attacks to be executed. The aim of this work is to crack as many passwords as possible

using the handshakes collected while walking through the metropolitan area of A Coruña. Additionally, using brute force, we built a targeted and contextualized dictionary that takes into account both the geographical location and the nature of the access point's owner. Finally, we reduced the size of the dictionary using a contextualized grammar.

## The four-way *handshake* or why your neighbor sleeps more soundly

New vulnerabilities in the WPA/WPA2 protocols have been discovered each year, but none of them have compromised the confidentiality of the authentication key. The four-way *handshake* has proven effective in preserving the security of that part of the protocol—each time a client wants to connect to an access point, they exchange four messages to confirm that the know the authentication key. This key is "hashed", that is, a function has been applied to it that transforms the arbitrary size key (in WPA2, from eight to 63 characters) into another set of fixed size characters.

In cryptography, a desirable quality of a *hash* function is that it is not reversible in practice, *i.e.*, it is computationally not feasible to calculate the inverse function and retrieve the input from the *hash* function. This is why passwords were obtained exclusively based on the available time and computing capacity since the WPA/WPA2 protocol was extended. The process of "dehashing" a password often involves three steps:

1. Loading many terms, combinations of terms, or combinations of terms and symbols to build a word.
2. Applying the *hash* algorithm to the word.
3. Comparing the output of the previous algorithm with the *hash* to be deciphered.

Therefore, on article, the only threat to the confidentiality of the WPA/WPA2 key is technological progress. The greater the computing capacity of the devices and the faster the input/output becomes, the more keys that can be tested in a reasonable time.

As is the case with the authentication approach outlined in *Vorakulpipat, Pichetjamroen & Rattanalerdnusorn (2021)*, potential alternatives are currently being researched. A case study of the implementation of a secure time attendance system, offers a security scheme that seamlessly integrates all traditional authentication factors plus a location factor into a single system. Comprehensive-factor authentication, which makes use of all available authentication factors, may maintain the necessary security level and usability in practical application.

Utilizing reputation and reputational models to assign a trust value to networked devices is another line of defense (*Fremantle & Scott, 2017*). Reputation is a basic idea that is utilized frequently in all fields of knowledge, including computer sciences as well as the humanities, arts, and social sciences. Reputation is used to gauge a system's trustworthiness in computer systems.

In light of this, the System-Theoretic Process Analysis for Security (STPA-Sec) (*Yu, Wagner & Luo, 2021*) method considers losses to be the outcome of interactions, focuses on reducing system vulnerabilities rather than preventing external threats, and is relevant to complex socio-technical systems. However, the STPA-Sec lacks effective advice for

defining information security concepts and pays less attention to concerns relating to information security than safety (such as data confidentiality).

Any of the approaches discussed above may be possible improvements or even solutions, but when authentication relies directly or indirectly on the existence of a password, the human factor is decisive. As will be seen in this study, the generation of passwords by users and administrators continues to be inefficient, causing password generation grammars to be extrapolated even today with a relatively high success rate.

### You promised me that 10 characters would be enough

The only threat to the confidentiality of the WPA/WPA2 key is technological progress. On article, with each new access data leak (such as LinkedIn, "Collection #1" or IO Verifications), you learn more about choosing a password. The ethical implications of using this type of leak for research are the subject of discussion (*Egelman et al., 2012*; *Thomas et al., 2017*). There is a tendency to use short, conspicuous passwords with combinations of numbers, letters, or words that are easy to remember. When web pages (or other systems) require a special number or character to be included, it is added to the end of the password (*WPEngine, 2021*; *Riley, 2006*), and since it is assumed that a secure password has already been created, partly because it is unknown to others, it is reused in other services (*Google, 2019*), increasing the likelihood of it appearing in some filter.

At best, when these leaks occur, the published passwords are "hashed". A "hashed" password is more secure than a plain-text password, but there are also better *hash* functions than others, and once a *hash* is made public on the Internet, anyone can try to crack it. Because the computing is distributed over many nodes, the time to decipher is considerably reduced and soon these collections become partly or completely plain text, as well as public. In the worst case scenario, companies save passwords directly in plain text and none of the above is necessary. The information provided by these leaks is especially valuable because they are real passwords, which may be used in statistical analyses and to compile huge dictionaries. In short, the "vulnerability" of WPA/WPA2 passwords depends on the person choosing them.

## STATE OF THE ART

To better understand the concepts used in this work, it is worth dedicating a few pages to explaining what the four-way *handshake* is, why it was designed that way, and how tools like Hashcat execute the decoding. We also discuss below what can be understood by *password security* and why a significant percentage of passwords will almost always be cracked due to user behavior.

The following explains the behavior of the four-way *handshake* from two processes informally known as "encryption" and "decryption". We first explain the calculation process, step by step, and clarify why the simplest options are not optimal from the point of view of password security, in order to build a clear picture of how an access point (router) and a client confirm they know the authentication key, privately and securely. Second, we explain what is calculated in the opposite process of decryption since, as you will see, it is not a question of undoing the steps of encryption one by one.

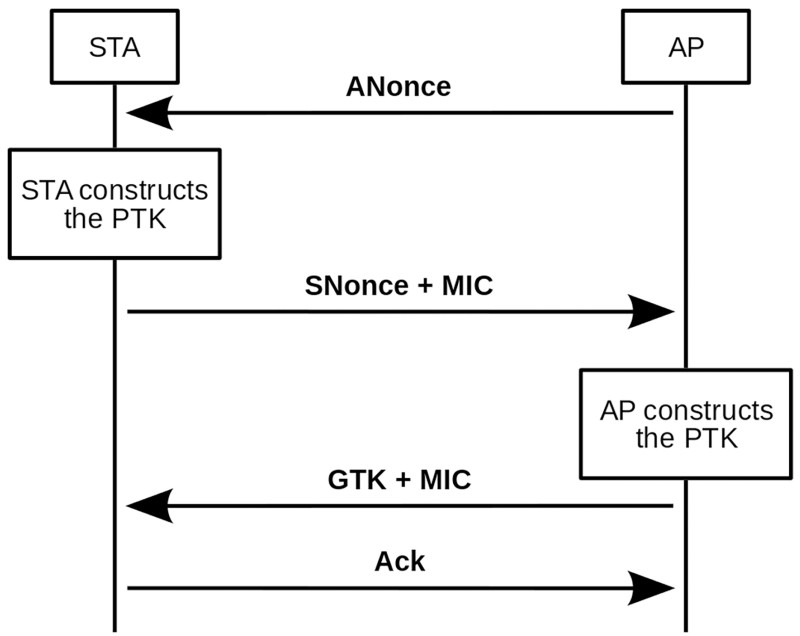

**Figure 1  Four-way *handshake* scheme.**

## The *handshake* process

The four-way *handshake* authentication process is explained below. In addition, Fig. 1 shows a scheme of this authentication process.

The basic principle behind the four-way *handshake* is that the pre-shared key (PSK) cannot be transmitted to the public medium under any circumstances. Therefore, the first thing the client should do before starting the message exchange is to build the pairwise master key (PMK), as shown in Formula (1).

$$PMK = PBKDF2(HMAC - SHA1, PSK, SSID, 4096, 256) \tag{1}$$

This is the version used by the WPA and WPA2 protocols. In order to understand it and the reasons for its choice, it would be interesting to shell it and build it step-by-step. One could start with a very simple version of this function (Formula (2)).

$$PMK = HMAC - SHA1(PSK) \tag{2}$$

HMAC-SHA1 is a *hashing* function and the PSK is used as input. This process returns a finite string of characters, the *summary*, which is inconsistent in the eyes of a human. This string could be used as a PMK, but this approach has a serious threat: *hash* tables. The result of entering two equal words in the HMAC-SHA1 function is the same, and is the same as saying that two equal passwords generate the same *hash*. The attack vector is clear, generates many of them with different passwords, and creates a *hash* table. These tables are distributed over the Internet and the only operation used to decipher a *hash* is to find it in this collection.

It is therefore essential to use this pseudo-random function (see Formula (3)) that prevents two identical passwords from generating the same *hash*. The result of applying a

**Table 1 PTK key decomposition.**

| PTK | | | | |
|---|---|---|---|---|
| 16 bytes | 16 bytes | 16 bytes | 8 bytes | 8 bytes |
| KCK | KEK | TK | MIC-tx | MIC-rx |

pseudo-random function is not completely random, as one can guess from its name, since it is determined by a set of initial values. The set of initial values in the WPA/WPA2 protocol is the SSID, *i.e.*, in order the name of the access point. This process is known as *salt* or "salting".

$$PMK = FUNCION\_PSEUDOALEATORIA(HMAC - SHA1, PSK, SSID) \qquad (3)$$

Although this version is better than the initial one, one can still add complexity to it. It is necessary to design a function that repeats this process a certain number of times to generate a key of a certain number of bits (see Formula (4)). This function is PBKDF2 and when it repeats the application of the pseudo-random function, 4,096 times, it generates a 256-bits key, the PMK.

$$PMK = PBKDF2(HMAC - SHA1, PSK, SSID, 4096, 256) \qquad (4)$$

From this function, one should be able to receive the first *handshake* message from the access point, the ANonce. With ANonce the client has all the necessary information to build the PTK, and it is only necessary to concatenate PMK, ANonce, Authenticator Mac Address (AMA), and Supplicant Mac Address (SMA). The result of the concatenation is passed as input to another pseudo-random function that will return a 64-byte (512-bit) string, the PTK (see Formula (5)).

$$PTK = PRF512(CONCAT(PMK, ANonce, SNonce, AMA, SMA)) \qquad (5)$$

The PTK is divided into five parts as shown in Table 1.

At this point, the client sends the second *handshake* message, composed of its SNonce and an Message Integrity Code (MIC). As the customer already has the PTK, in addition to sending the SNonce, it protects it with an MIC. The MIC is calculated from the SNonce and the Key Confirmation Key (KCK). WPA and WPA2 use different *hash* functions for this step, as can be seen in Formulas (6) and (7), respectively.

$$WPAMIC = HMAC(KCK, SNonce, MD5SHA) \qquad (6)$$
$$WPA2MIC = HMAC(KCK, SNonce, SHA1) \qquad (7)$$

Thus, when the access point receives SNonce and MIC, it can also calculate the PTK (until now it lacked SNonce) and with SNonce and PTK, it can validate the MIC (that is, calculate it and compare if it is the same as the one it has received). If the MIC validation is positive, the access point confirms that it and the client are using the same PTK, which was the initial target. At this point in the *handshake*, the person intercepting the messages has enough information to make an attempt to decipher the password. Messages 3 and 4 of the

process are not relevant to the purpose of this article, so they will be ignored in the following explanations.

### Decoding

Once the second *handshake* message is intercepted, only the PSK is missing when reconstructing the PTK. A PSK must be assumed to perform the computation process. It is at this point that a dictionary, brute-force attack, or another method of word generation is used. For each word, the corresponding PTK is generated, from which the KCK is extracted and the MIC is calculated. If the MIC matches the genuine MIC, then the password has been found.

Naturally, each test with each word is totally independent from the others and no communication between nodes is necessary, which allows for the perfect parallelization of the decoding process. This is a case where it is preferable to have many low-power processors rather than some high-power ones. Graphics cards fit this description.

## THE SECURITY OF PASSWORDS

There are numerous studies on the suitability of passwords as a form of system authentication, not only from a technical computing perspective but also from the psychological perspective of users. In 1979, the security of passwords in UNIX systems was explored by *Morris & Thompson (2002)*.

Passwords are commonly used to represent system design because they have a usually simple implentation that is easy to understand for whoever uses the system. As mentioned above, users are often the weakest link in the chain and compromise the security of systems by forgetting passwords, writing them down, sharing them with others, or defining them in easy-to-guess patterns. The extent to which passwords are weak due to a person's lack of motivation or cognitive limitations remains an open question. Some studies conclude that there is not yet strong evidence that people who are aware of how to enhance password security actually choose passwords that prevent cracking by a knowledgeable attacker, as they are often considered cumbersome or impractical (*Bonneau, 2012*; *Notoatmodjo & Thomborson, 2009*). Attempts to educate the "average user" about creating more secure passwords through advice and enforcement of password policies are often unsuccessful. Password requirements are minimally met or the advice given for their creation is ignored or misunderstood (*Forget et al., 2008*). For example, a policy that requires including at least three digits in a password will often result in the user simply adding "123" to the end of an insecure password. *Cracking* tools include large sets of rules like those (*Weir et al., 2010*).

Other works found little difference between normal users and "conscious" users due to the fact that most tend to vary the complexity of passwords as the importance of the accounts they want to protect varies. Most people who participate in studies have a similar sense of the trade-off between security and convenience when choosing a password and understand that reusing passwords or using weak passwords for their important accounts could put their data at risk (*Notoatmodjo & Thomborson, 2009*; *Wash et al., 2016*; *Gehringer, 2002*).

While the technical computer approach or, in other words, *what makes a password secure*, does apply in the specific case of Wi-Fi passwords, the results of the psychological approach in general do not seem to be extrapolated. First, because in most of the works reviewed (*Riley, 2006*; *Notoatmodjo & Thomborson, 2009*; *Forget et al., 2008*; *Bryant & Campbell, 2006*; *Yan et al., 2004*; *Komanduri et al., 2011*; *Narayanan & Shmatikov, 2005*) the psychological processes of choosing a password (both when creating and reusing another one) imply that *the system* asks for a password. This is radically different to Wi-Fi routers where, by default, the password is fixed and it is *the user* who voluntarily changes it. Secondly, the Wi-Fi password only needs to be entered once per device, with the operating system being responsible for keeping it secure for each new authentication.

The most popular strategies for reducing the weaknesses inherent in password use include the following (*Bryant & Campbell, 2006*):

1. Password lengths of at least eight characters: longer passwords increase the time it takes for "cracking" programs to decrypt them.
2. Randomly combining uppercase and lowercase letters with special symbols: including uppercase, lowercase and symbols (!£$%, *etc.*) in passwords requires the use of brute force methods and increases the number of character permutations to be tested. A proposed way to approach this strategy is to use mnemonic rules (*e.g.*, "I want to generate a secure password against cracking Software" generates "IwtgaspacS" by taking the first letter of each word).
3. Not using dictionary words to minimize dictionary attacks.
4. Changing the password periodically: If an intruder obtains a valid password, most systems allow them to continue accessing until the intrusion is detected. If users periodically change their passwords, the intruder will be forced to identify the new password.

Regarding strategy number 2, users rarely generate random passwords (a perfectly randomly generated set of passwords would produce an even distribution of characters). Cracking software optimizes brute-force attacks that use character frequency tables by exploiting an inherent property of language: certain characters appear more often than others. According to *Narayanan & Shmatikov (2005)*, when users are invited to create passwords based on mnemonic standards, most select lyric phrases from songs, movies, literature, or TV shows. The text from these sources is often available on the Internet. This opens up the possibility of building an effective dictionary for these passwords as well.

Strategy 4 has been considered for several years to be a generally incorrect recommendation (*Adams & Sasse, 1999*; *Mazurek et al., 2013*). Passwords should be changed when you have the intuition or knowledge that they have been compromised or when they have become potentially insecure due to technological advancements.

These strategies make creating passwords and memorizing them a particularly difficult task, especially since the proliferation of password management services on the Internet means that all passwords have to be managed but none can be reused. However, this is a problem that, as mentioned above, does not affect Wi-Fi passwords since *they only need to*

*be entered once*. With this difficulty overcome, there is no excuse for not using long and complex passwords selected entirely at random. Knowing that these requirements are contrary to the properties of human memory (*Yan et al., 2004*), several security experts have argued that writing down the password and keeping it in a safe place could be a good option for the average user (*Schneier, 2005*). It is not surprising then that different studies a relationship between writing down passwords and making them more secure (*Komanduri et al., 2011*).

A long and complex password is required for the PSK of the WPA/WPA2 protocol, since "cracking" the *handshakes* is done *offline*. The key issue is to answer the questions of "how complex" and "how long it must be", for which there is no single, sure answer. If we assume that we want to generate a password that can only be encrypted through brute force attacks while avoiding dictionary words, we can think of the following practical approach.

$$VR_{x,y} = x^y \tag{8}$$

The password search space will be the variation with repetition of $x$ order elements $y$ (see Formula (8)), where $x$ is the cardinality of the character set to be used and $y$ is the length of the word. The cardinality is limited by the characters allowed by the WPA/WPA2 protocol, the printable ASCII characters: digits (10 characters), uppercase letters (26 characters), lowercase letters (26 characters), and symbols (33 characters). Arguably, the simplest and most effective way to improve password security for the average user is to increase its length. Increasing it produces an exponential growth of the search space, while expanding its complexity by increasing the cardinality of the character set produces only a polynomial growth, which has been empirically proven. Growing in length also allows the user to not use special characters, which can cause problems with poor implementations of the WPA/WPA2 protocol. The following two password composition policies should result in passwords with the same entropy: one that requires passwords to be at least 16 characters long and another that requires at least eight characters, but also a capital letter, a number, a symbol, and a dictionary check, according to the best available guidelines (*Burr, Dodson & Polk, 2006*). However, the 16-character policy produces significantly less predictable passwords and, according to several metrics, is less costly for users (*Komanduri et al., 2011*).

In short, the two attack vectors that are exploited in this work to decipher the intercepted Wi-Fi passwords are: first, to rely on the existence of passwords in a search space that can be covered by brute force with the available infrastructure and time; and, second, to rely on human carelessness that has led to the use of typical passwords that were subject to some kind of leak. The average success of these approaches is usually around 30% in mass decryption projects, while published numbers of various studies on password cracking effectiveness vary substantially: gpuhash.me (28.70%), and wpa-sec (31.93%). According to *Bonneau (2012)*:

- Most studies have deciphered 20–50% of dictionary-size passwords in the $2^{20}-2^{30}$ word range.

- All studies show decreasing yields for larger dictionaries.
- There is little data on the efficiency of small dictionaries, as most studies use the largest dictionary they can process.

## Previous works on analyzing password patterns

*Bonneau (2012)* investigated how people form passwords by looking at many publicly available password sets, such as RockYou and CSDN. They defined a variety of password properties, including all digits, non-ascii characters, and the usage of neighboring keys. The article used a pattern of adjacent keys excluding repetitions as an indicator of a keyboard pattern, and found that this pattern appeared in 3% of RockYou passwords and 11% of CSDN passwords.

*Veras, Collins & Thorpe (2014)* proposed an intriguing paradigm for categorizing passwords meaningfully. To segment passwords, they produced all feasible segmentations of a password and then used a source *corpus* to find the ones with the best coverage (which we would refer to as a training dictionary).

In many other studies, dictionaries were utilized to generate variant guesses by applying mangling criteria as well as as a source of passwords. For some early work in this area, see *Klein (1990)*. An article by *Dell'Amico, Michiardi & Roudier (2010)* is an example of recent work. This research took into account a number of dictionaries accessible from John the Ripper, and an assessment was made by comparing the passwords cracked using only the dictionary entries. Two conclusions emerged: it is preferable to employ the same kind of dictionary as the target type (for example, Finnish when attacking Finnish passwords), and while bigger dictionaries are better, they have decreasing benefits.

Using a Markov model (*Narayanan & Shmatikov, 2005*), *Castelluccia et al. (2013)* aimed to use personal information to improve password cracking. Their OMEN+ technology combined personal information with standard training based on passwords that have been disclosed.

*Castelluccia et al. (2013)* looked into the usage of personal data like email, birthdays, and usernames, but they did not utilize prior passwords or cross-site passwords. *Castelluccia et al. (2013)* tweaked the 3-gram probabilities so that a complete test set yielded better results. The suggested method, on the other hand, employed a very efficient PCFG-based training system that avoided the drawbacks of the Markov technique. Furthermore, it did not need any changes to the training or cracking components; instead, it merely necessitated the creation of extra grammars and dictionaries.

Using a 12,306 obtained from a Chinese train ticket website, *Li, Wang & Sun (2016)* looked at the usage of personal information in password cracking. To build the Personal-PCFG system, *Li, Wang & Sun (2016)* enhanced *Weir et al. (2009)* probabilistic's context-free grammar method. *Wang et al. (2016)* also investigated the usage of prior passwords and enhanced probabilistic context-free grammars (*Weir et al., 2009*). They used a tagged variable system instead of a new variable B for personal information such as birthdays, with the subscript denoting length, as *Li, Wang & Sun (2016)* did.

**Table 2 Brute force password calculation.**

| Password | Search space | Time to go through space |
|---|---|---|
| 1111111111 3548346841 | 10,000,000,000 | About 7 h |
| Holahola12 83dk12w7p1 | 2,758,547,353,515,625 | About 220 years |
| HolaHola12 83Dk12w7P1 | 604,661,760,000,000,000 | About 47,934 years |
| HolaHola1# 8#Dk1^w7P1 | 38,941,611,811,810,745,401 | About 3,087,076 years |

*Das et al. (2014)* evaluated data from publicly accessible leaked password sets with user IDs to determine passwords for the users. They were able to uncover 6,077 distinct individuals with at most two passwords apiece; 43% of the passwords were the same, while the rest were non-identical. *Das et al. (2014)*, on the other hand, did not take into consideration the modifications that a user may make while using identical passwords for the same or different accounts. *Zhang, Monrose & Reiter (2010)* carried out a large-scale investigation on password changes prompted by password expiration. They were able to get access to a database of over 7,700 accounts, each with a known password and a password that was later changed. They simulated a password change as a series of steps.

## The art of "cracking" passwords

Some of the concepts used throughout this work are shown below with simple examples.

In Table 2, we took 10-character passwords and using a current graphics card, the NVIDIA GTX1080, as a reference, were able to calculate around 400,000 *hashes* per second (*Gosney, 2016*). The search space corresponds to the number of possibilities for each combination of ASCII characters: numeric only, alphanumeric with lower case letters, alphanumeric combining upper and lower-case letters (mixed), and mixed alphanumeric with special symbols. The time to go through the space was the division between the space and the search and the power of the graphic card, or in other words, the time it would take to decipher the password if it was the last one to try the entire search space, the worst case scenario.

These times correspond to what are called brute-force attacks. The only variables that require a brute-force attack to be executed are the bearable characters and the length of what is attempted to be deciphered. That is why, for this type of attack, there is no distinction between "holahola12" and "83dk12w7p1" when anyone, at first glance, can intuit that one *must be* easier to decipher than the other.

That intuition is what the word collections or dictionaries condense. It is known that the password "holahola12" is statistically more likely than "83dk12w7p1", not only because "hello" is a word frequently used in the Spanish language, but also because it has been found in the real world by other people who have already used it. According to Pwned Passwords (https://haveibeenpwned.com/Passwords) "holahola12" was found in

721 leaks. A brute-force attack that knows the two variables mentioned in the previous paragraph will, in time, always crack the password. The problem is that many times one neither knows the characters used, the length of the word, or they do not have the necessary time, and that is why a compromise is sought: the search space is restricted to a subset in exchange for reducing the chances of success by deciphering the key.

The easiest option is to use one of the most popular dictionaries in circulation on the Internet, which range in length from hundreds of thousands to billions of words. Many of them tend towards most Anglo-Saxon terms, but for this strategy based on pure probability, it is not a disadvantage. The other option is to develop one's own specialized dictionary when knowing some target data such as the country, the name of the owner of the access point, or the company's password length policy, and from that propose some patterns that generate the words of the dictionary.

For example, an audit is to be performed at a point in Spain and the pattern chosen is "two Spanish words plus one or two digits or special symbols". We would start by choosing the words that RAE publishes as the most used in Spanish (*RAE, 2021*), out of which the first 10,000 are taken ("hola" is number 7,373) and those with fewer than four and more than eight characters are discarded, leaving 5,935, and each one is concatenated with the other 5,934 and with itself. This generates 35,224,225 base words. Adding one digit or special character generates 1,549,865,900, and adding two digits or special characters generates 68,194,099,600. Finally, there is a total of 69,743,965,500 words, which the reference graphics card can be tested in two days in the worst case.

The two strategies can and do complement each other in this work. A first sweep with generic dictionaries must decipher a sufficient amount of passwords to be able to infer a series of patterns with which to generate a minimum specific dictionary for the metropolitan area of A Coruña. Thus, on the one hand, new and more specialized candidate words are available and, on the other hand, it would not be necessary to use high-performance hardware to crack the passwords in the area.

## MATERIALS AND METHODS

This section is focused on Hashcat (a *hash-cracking* program), the infrastructure on which this entire process was executed, and the strategy behind the decisions to use different types of attacks. Afterwards, with the already deciphered passwords, we will comment on the patterns that were found and how a minimum dictionary was elaborated for the metropolitan area of A Coruña.

This section is dedicated to the decryption of passwords and focuses on the three essential questions: the software used to decode, the hardware on which it runs, and the strategy to follow, given that both time and computer resources are limited.

### Hashcat

Hashcat (version 4.2.1) was chosen to decipher the *handshakes*. Hashcat is an open source password recovery tool that supports five attack modes for about 200 hashing algorithms for both CPU or GPU and other hardware accelerators on Linux, Windows, and MacOS.

## Infrastructure

For the password cracking phase, access to two NVIDIA Tesla K80 and two other NVIDIA Tesla K20m graphics cards was provided. The Tesla K80 features 4992 CUDA cores, 24 GB of GDDR5 memory, and 480 GB/s of aggregate memory bandwidth. The Tesla K20m has 2496 CUDA cores, 5 GB of GDDR5 memory, and 206 GB/s of aggregate memory bandwidth.

## Strategy

The more time and resources available to crack a password, the less useful the strategy is. It simply feeds the tool of choice with the various intended and expected attacks. However, in an environment where time or resources (or both) are limited, it is logical to lead the first efforts towards the more obvious operations in an attempt to achieve results in a short period of time. The strategy can be divided into three phases: obvious overlaps, mass tests, and common variations of base words.

It is appropriate to make two observations on the estimates that will follow. The first is that, in order to make estimates comparable between the different attacks, the entire set of *handshakes* has always been considered. In practice, as they were deciphered, they were not tested again. The second is that it should be taken into account that dictionary attacks with rules are faster than dictionary attacks due to the PCI-e bandwidth limit and latency in the transfer of information from the host. That is, if one has a base word dictionary and rule set, it is much more time-efficient to send only the base words to GPU and have them modified on the GPU itself than to generate the entire collection of modifications on CPU and then send them to the GPU.

### First phase

The first attempts were two dictionary attacks using PasswordsPro and RockYou, two of the most known and used word collections. PasswordsPro is a collection of words made by the InsidePro team while RockYou is the result of a 2009 leak of access data from RockYou, a company that developed widgets for MySpace and applications for other social networks such as Facebook. RockYou used an unencrypted database to store personal information and passwords and was the target of an SQL injection attack.

A widespread practice is to use the DNI or "Spanish ID" as a password since it is eight digits and one letter, thus meeting the minimum requirements of some registration forms. The general idea in the Spanish population is that the letter is random, which is incorrect; it is calculated from the digits. Therefore, what could be a search space of $36^9$ terms, effectively becomes an order of magnitude smaller ($10^8$).

The next two were brute-force attacks with eight- and nine-digit numbers. The eight- and nine-digit search space has $10^8$ and $10^9$ of possibilities, respectively.

Table 3 shows the times consumed by all the attempts to solve the 133 targets at 937,400 H/s (maximum capacity of the infrastructure used) and their respective results:

It was then tested with the names of the access points themselves along with a set of rules provided by Hashcat called "best64.rule", Table 4. Some examples of the modifications made to the base words were: turning it over, putting it in capital letters, and

**Table 3 Passwords decrypted and time consumed by all attempts using only dictionaries.**

| Dictionary | Words | Time consumed | Decrypted |
|---|---|---|---|
| PasswordsPro | 2,937,125 | 6 min | 0 |
| RockYou | 14,344,391 | 33 min | 13 |
| DNI | 100,000,000 | 236 min | 0 |
| 8 Digits | 100,000,000 | 236 min | 11 |
| 9 Digits | 1,000,000,000 | 2,364 min | 2 |

**Table 4 Passwords decrypted and time consumed by all attempts using dictionaries and applying rules.**

| Dictionary | Rules | Words | Time consumed | Decrypted |
|---|---|---|---|---|
| ESSIDs | best64.rule | 1,432,354 | 3 min | 2 |
| Names | best64.rule | 2,017,400 | 4 min | 1 |

**Table 5 Password decrypted and time consumed by all attempts using dictionaries and the "best64.rule" rule set in second phase.**

| Dictionary | Words | Time consumed | Decrypted |
|---|---|---|---|
| Hk_hlm_founds | 38,647,798 | 91 min | 3 |
| CrackStation | 63,941,069 | 151 min | 0 |
| Top306M ProbableWordlists | 306,429,377 | 724 min | 4 |
| HashesOrg | 446,426,204 | 1,055 min | 0 |
| Andronicus | 626,198,124 | 1,500 min | 0 |
| Breachcompilation | 1,012,024,699 | 2,395 min | 0 |
| Rocktastic | 1,133,849,621 | 2,681 min | 0 |
| Weakpass_2_wifi | 2,347,498,477 | 5,551 min | 5 |
| Weakpass_2 | 2,649,982,129 | 6,266 min | 0 |

adding a number at the end or at the beginning. In this case, first the ESSIDs of the audited access points were used as a dictionary and then the 100 most common names of men and women in Spain according to the INE (*INE, 2018*).

In these two cases, the accelerating effects of the rules were not relevant given the minimal amount of ESSIDs (131) and names (200).

### Second phase

The second phase tested some of the smaller dictionaries with the "best64.rule" rule set. It is in these cases that one can notice the acceleration of the use of rulers as well as the work that Hashcat does by rejecting certain words according to its preconfigured policies. Following these policies, up to 90% of all candidate words can be rejected. Therefore, although "best64.rule" produces 10,087 variations, it does not translate into 10,087 times more "cracking" time, Table 5.

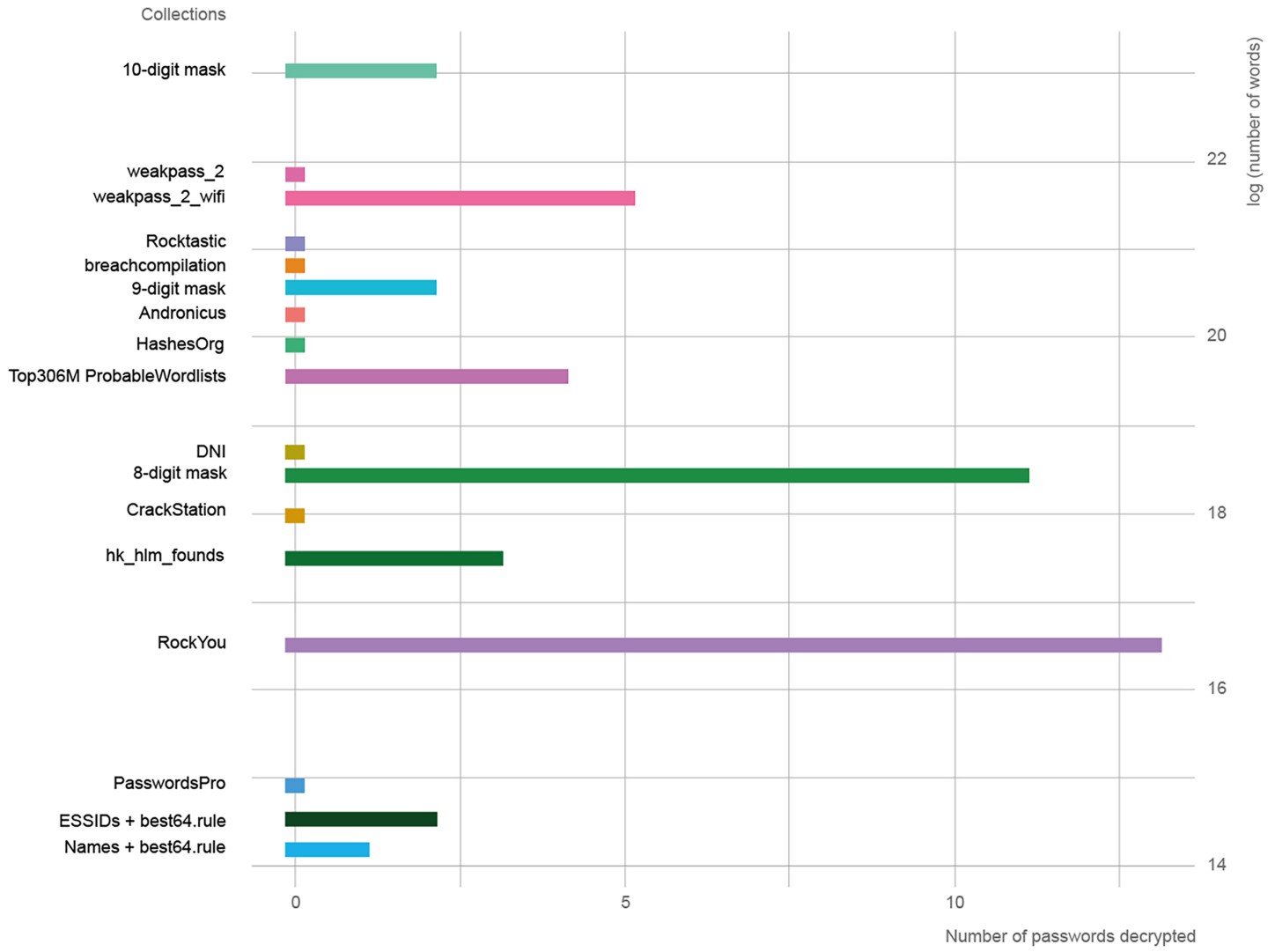

**Figure 2** Relationship between the length of word collections and decoded passwords.

In Fig. 2, the length of the collections is represented in logarithmic scale against the success rate of each one. The three collections with the best results were RockYou, 8-digit mask, and weakpass_2_wifi. It is necessary to use the natural logarithm given the large variation in length between collections, since there are four orders of magnitude from the smallest to the largest.

The final number of decoded passwords amounted to 44, a 33.08% success rate, which corresponded with other empirical evidence already mentioned. Perhaps the most worrying fact is that, following this strategy, in less than two days of computing, 22.13% of passwords had already been recovered, which was approximately two thirds of the total that would eventually be cracked. As shown in Figs. 3 and 4, passwords of 10 characters or less were dominant in length and in type, numerical, and alphabetical passwords. This data

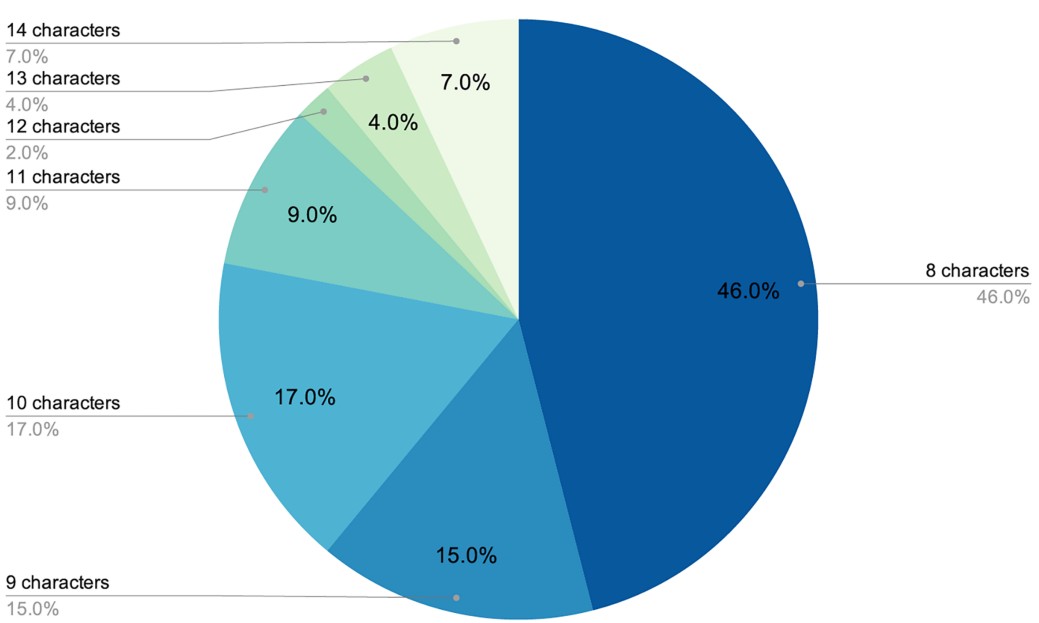

**Figure 3  Relationship between the size of word collections and decoded passwords.**

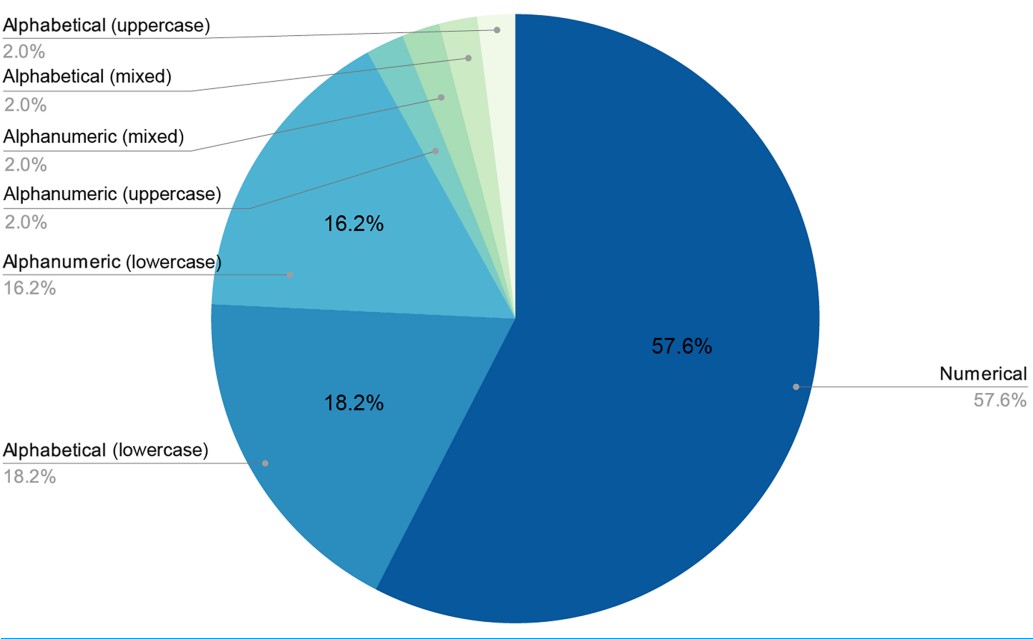

**Figure 4  Length distribution of decoded passwords.**

is not strange, since the higher the length or complexity of the passwords, the more difficult it is to crack them.

## Handshakes collection

The area for collecting the handshakes covers an area of about 105,785 $m^2$ and a perimeter of about 1,518 $m$ (see Fig. 5). This is an interesting area because of the presence of all kinds

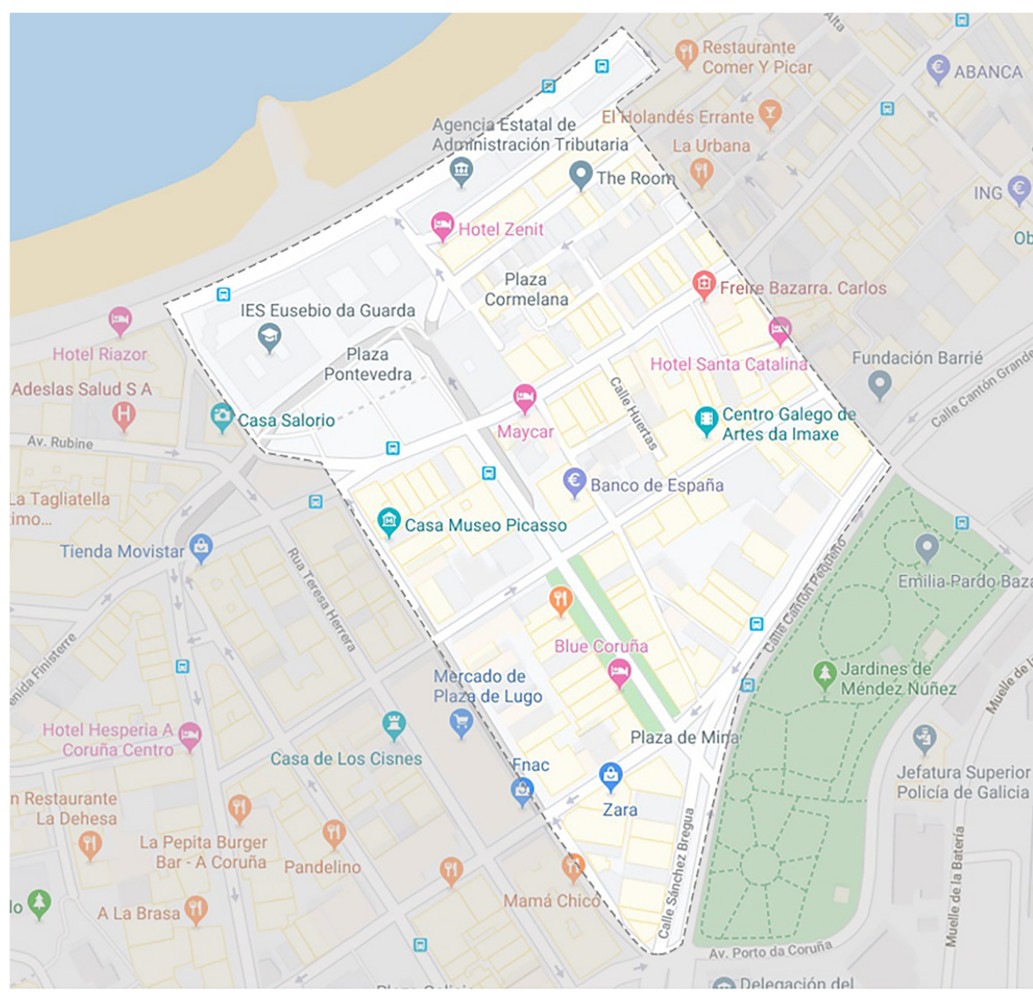

Map data © 2022 Inst. Geogr. Nacional, Google.

**Figure 5 Map of the audit area.** Map data ©2022 Inst. Geogr. Nacional, Google.

of homes and businesses, including banks, small shops, pharmacies, restaurants, and bars. It has been considered a sufficiently representative area of the metropolitan area of A Coruña.

It is also interesting to know the concentration of dwellings and residents (in main houses) due to technical issues: to capture a *handshake* it is necessary to have devices connected to the access point, which in most cases means that the dwellings are occupied by people using Wi-Fi devices. According to the latest INE census data, the chosen area covers the census sections that shown in Table 6 in large part or in full.

As shown, the population concentration is not particularly high, 2.14 people per house, a fact that makes it difficult to obtain *handshakes a priori*.

The audit to obtain the handshakes lasted 19 h distributed over 11 days, between May 21 and June 11, 2019, during which about 15 kilometers were covered. The number of *handshakes* that was set as a goal to achieve was 100–200 in order to ensure a reasonable number of decrypted passwords. Initially, no method was followed to go through the area.

**Table 6 Census data.**

| Census tract | Residents in main dwellings | Dwellings |
|---|---|---|
| 15-030-01-004 | 1,590 | 775 |
| 15-030-01-012 | 975 | 475 |
| 15-030-03-001 | 780 | 310 |
| Total | 3,345 | 1,560 |

Once the number of *handshakes* exceeded 100, the locations were no longer random, but rather we attempted to go to those locations where none had yet been intercepted. The numbers that represent the results of this massive audit are as follows: 333 individual audits were carried out in which 239 *handshakes* were obtained, 142 were unique, therefore a 42.64% probability of interception in each audit. Of these 142, 133 corresponded to the PSK authentication and nine to the RADIUS server authentication. Consequently, the final number of *handshakes* to be deciphered was 133, equivalent to one for every 795 $m^2$.

## PATTERN ANALYSIS AND MINIMUM DICTIONARY DEVELOPMENT

A quick visual examination of cracked passwords revealed a pattern underlying the vast majority. To describe it we define two sets of words: base words and endings. If a word is in a set of base words $S_1$ (common name, business name, or common words of the language) and a set of endings $S_2$ (numbers, days, months, or years), a word may belong to the dictionary if it is in the next set. This approach is shown in Formula (9).

$$D = \{v_1 \ldots v_n t / v \in S_1,\ t \in S_2,\ v_1 \neq \lambda,\ 8 \leq |v_1 \ldots v_n t| \leq 63\} \tag{9}$$

That is, the set of all those word concatenations belonging to the sets $S_1$ and $S_2$ in which the only word that cannot be the null strip ($\lambda$) is the first one ($v_1$) and provided that the resulting concatenation has a length between eight and 63 characters.

The union of the following sets forms the set $S_1$ (see Formula (10)) of base words:

$B_{lambda} = \{\lambda\}$

$B_{nat} = \mathbb{N}$

$B_{names} =$ Set of the 100 most frequent names in Spain according to the INE

$B_{business} =$ Random subset of A Coruña business names

$B_{words} =$ Subset of the most used nouns according to RAE data

$$S_1 = B_{lambda} \cup B_{nat} \cup B_{names} \cup B_{business} \cup B_{words} \tag{10}$$

The names of persons were obtained from the list of the 100 most common male and female names according to the INE. Three variations were applied to those 200 names: all letters in upper case, all in lower case, and only the first one capitalized. The business names were obtained by *scraping* the Google Maps platform and a manual cleaning was performed (elimination of spaces, or invalid characters) to make them suitable.

On the other hand, the $S_2$ set of terminations is the merger of the following natural subsets (see Formula (11)):

$T_{num} = \{u/u \in \mathbb{N}, 0 \leq u \leq 9\}$

$T_{dia} = \{u/u \in \mathbb{N}, 1 \leq u \leq 31\}$

$T_{mes} = \{u/u \in \mathbb{N}, 1 \leq u \leq 12\}$

$T_{ano1} = \{u/u \in \mathbb{N}, 0 \leq u \leq 20, 78 \leq u \leq 99\}$

$T_{ano2} = \{u/u \in \mathbb{N}, 1{,}978 \leq u \leq 1{,}999, 2{,}000 \leq u \leq 2{,}020\}$

$$S_2 = T_{num} \cup T_{dia} \cup T_{mes} \cup T_{ano1} \cup T_{ano2} \tag{11}$$

For the minimum dictionary it is necessary to limit the word patterns (see Formula (12)). The minimum dictionary is made up of the merger of the following sets of words:

$B_1 = \{w/w \in \mathbb{N}, 10{,}000{,}000 \leq w \leq 19{,}999{,}999\}$

$B_2 = \{w/w \in \mathbb{N}, 20{,}000{,}000 \leq w \leq 29{,}999{,}999\}$

$B_3 = \{w/w \in \mathbb{N}, 269{,}000{,}000 \leq w \leq 270{,}999{,}999\}$

$B_4 = \{w^2/w \in \mathbb{N}, 0 \leq w \leq 9{,}999\}$

$B_5 = \{w^2/w \in \mathbb{N}, 0 \leq w \leq 99{,}999\}$

$B_6 = \{w^2/w \in \mathbb{N}, 0 \leq w \leq 999{,}999\}$

$B_7 = \{uvw/1 \leq u \leq 31, 1 \leq v \leq 12, 1{,}978 \leq w \leq 2{,}020\}$

$B_8 = \{0123456789, 12345678, 123456789, 1234567890, 1223334444, 1122334455,$
$\qquad 135792468, 1357924680, 246813579, 2468013579, 12345678910,$
$\qquad 123456789123456789\}$

$B_9 = \{uv/u \in B_9, v \in \text{ASCII upper and lowercase set}\}$

$B_{10} = B_{business}$

$B_{11} = B_{business} \times T_{year2}$

$B_{12} = B_{business} \times T_{month} \times T_{year1}$

$B_{13} = B_{business} \times T_{num}$

$B_{14} = B_{13} \times T_{num}$

$B_{15} = B_{name}$

$B_{16} = B_{name} \times T_{ano2}$

$B_{17} = B_{name} \times T_{month} \times T_{year1}$

$B_{18} = B_{name} \times T_{num}$

$B_{19} = B_{18} \times T_{num}$

$B_{20} = \{vw/v \in B_{words}, w \in T_{year2}, 4 \leq |v| \leq 7\}$

$B_{21} = \{vw/v \in B_{words}, w \in B_8, 4 \leq |v| \leq 7\}$

$B_{22} = B_{words} \times B_{words}$

$B_{23} = B_{22} \times T_{num}$

$B_{24} = \{w/w \in B_{words}, |w| \geq 8\}$

$B_{25} = B_{24} \times T_{num}\}$

$B_{26} = B_{25} \times T_{num}\}$

$$D_{min} = \{u/u \in \bigcup_{i=1}^{26} B_i, 8 \leq |u| \leq 63\} \tag{12}$$

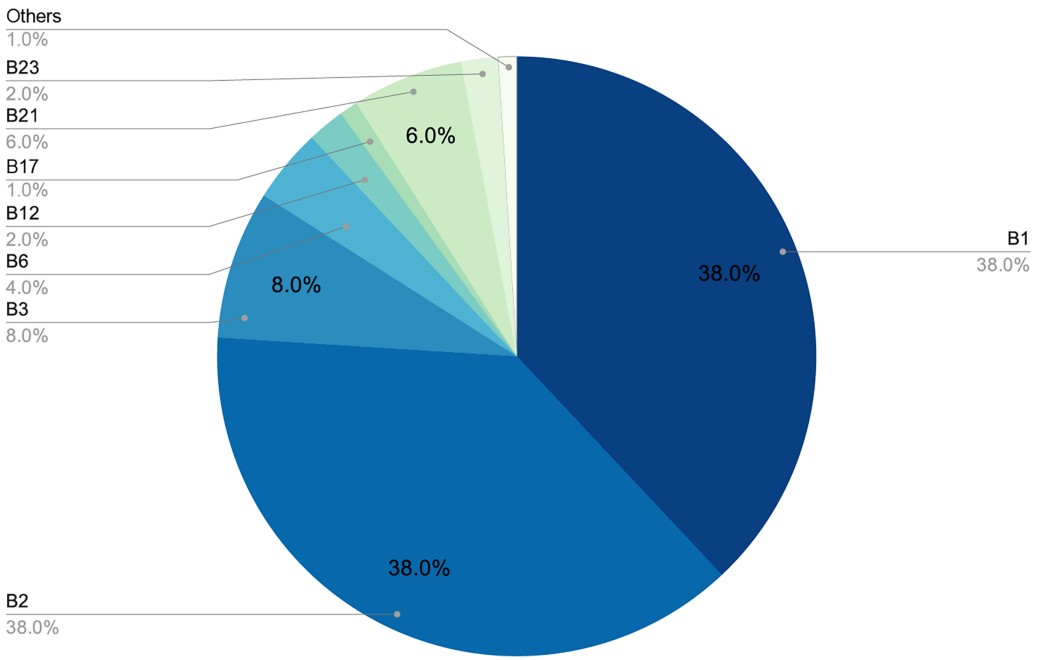

**Figure 6** **Proportion of each pattern in the dictionary.**

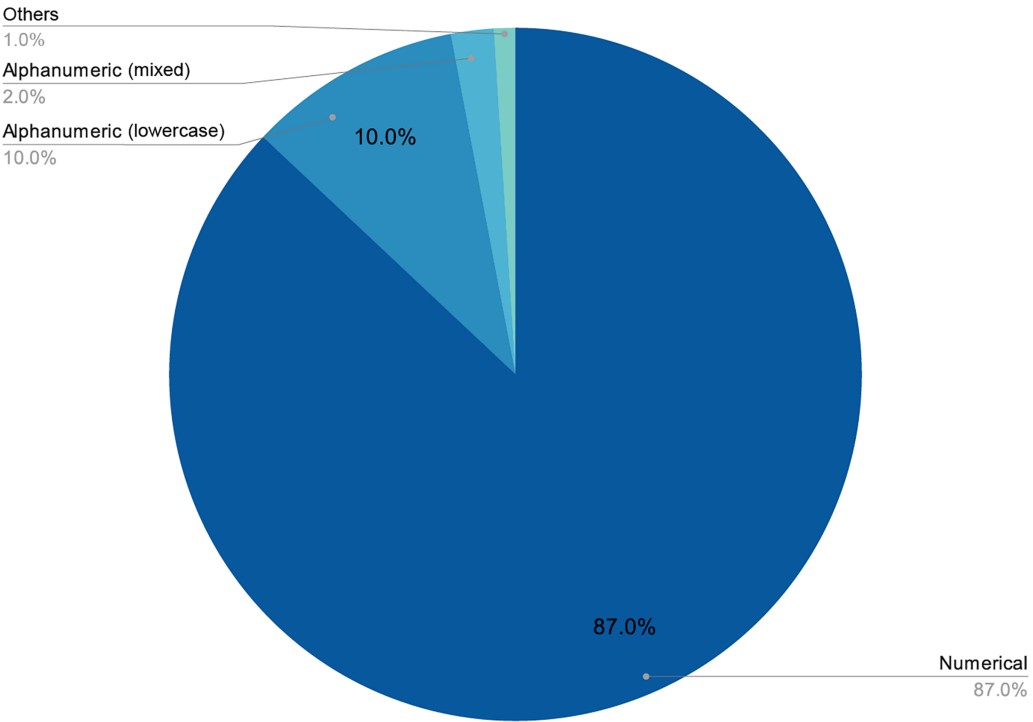

**Figure 7** **Proportion of each pattern in the dictionary.**

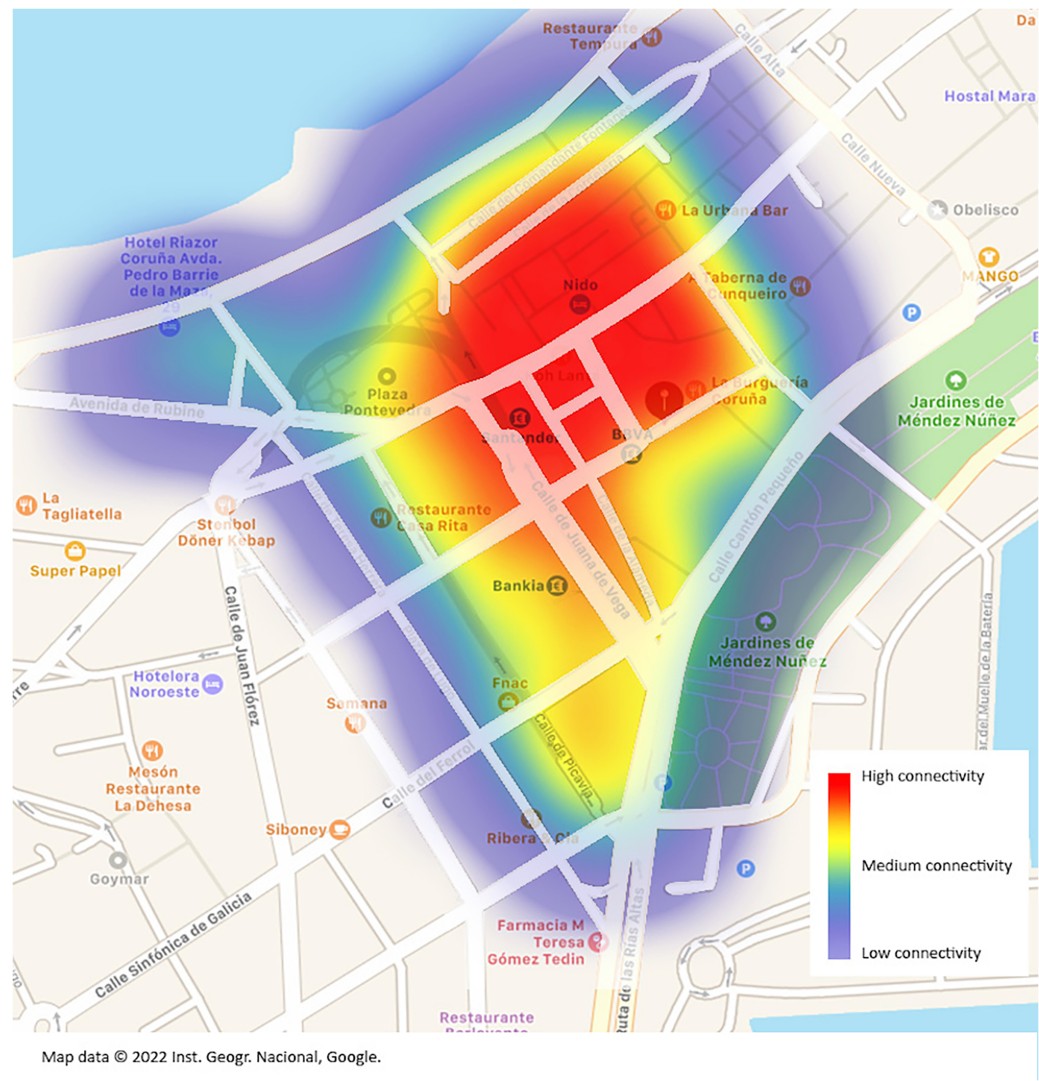

Map data © 2022 Inst. Geogr. Nacional, Google.

**Figure 8** **Heatmap of the routers form which the password has bees decoded.** Map data ©2022 Inst. Geogr. Nacional, Google.

Each pattern in the list generated a dictionary of words whose length should be between 8 and 63 (both inclusive) ASCII characters to meet the requirements of the WPA/WPA2 protocol. After joining all the sets, a single file of 288 megabytes was obtained.

Figures 6 and 7 provide a detailed visual overview of how the different subdictionaries are divided according to importance, as well as the types of words that make them up.

## CONCLUSION

As stated above, humans are lazy and predictable. Figures 4 and 5 show that our password patterns are still not very random. Even when having access to a superior set of symbols (special characters and distinction between upper and lower case), the collected patterns show a great predominance of numeric and lower case alphabetic ones. This predictability can be illustrated in an everyday example. When we are asked to create a password

nowadays (ignoring automated random password creation systems) we are asked to use "at least one special character and one uppercase letter". Again, Figs. 4 and 5 show that this "recommendation" is interpreted literally by the user, who includes "only" a special character and an uppercase letter. All this means that even today, dictionaries of this type can be created with a considerably high success rate. Given that this human behavior will continue password generation restrictions should ensure that dictionaries of this type will not be sufficient *a priori*.

Likewise, the context or destination of the access point used for the creation of your security password continues to give credibility to the adjectives used for human administrators.

Thanks to the equipment used, the location information for each audit can be saved, and a visual representation on a map can be produced showing the assumed range of each access point from which the password was decoded. It should be noted that the position where each access point was located is where *the audit* was executed. It is accepted that in each audit, we waited for the GPS to determine a position with a margin of error of 7 m or less because only targets with a good signal quality were set.

In Wi-Fi networks the range depends largely on the environment in which the router is deployed. We made a very basic estimate and did not consider any key elements such as signal obstacles, height (the access point may be on a high floor of a building), or router model. Heights can vary from 15 m inside a house to 100 m outdoors. For the sake of simplicity, the 15, 30, and 45 m radios were chosen to represent the quality of the signal. It can be observed that with one *handshake* per 795 $m^2$, and with a 30% average success rate in deciphering passwords, the audited geographical area can be completely covered (Fig. 8). It would be possible to go through the entire area by connecting to different routers without ever losing the Internet connection.

### Funding

This work is supported by the General Directorate of Culture, Education and University Management of Xunta de Galicia (Ref. ED431G/01, ED431D 2017/16), the Galician Network for Colorectal Cancer Research (Ref. ED431D 2017/23), Competitive Reference Groups (Ref. ED431C 2018/49) and the Spanish Ministry of Economy and Competitiveness *via* funding of the unique installation BIOCAI (UNLC08-1E-002, UNLC13-13-3503) and the European Regional Development Funds (FEDER). The funders had no role in study design, data collection and analysis, decision to publish, or preparation of the manuscript.

### Grant Disclosures

The following grant information was disclosed by the authors:
General Directorate of Culture, Education and University Management of Xunta de Galicia: ED431G/01, ED431D 2017/16.
Galician Network for Colorectal Cancer Research: ED431D 2017/23.

Competitive Reference Groups: ED431C 2018/49.
Spanish Ministry of Economy and Competitiveness *via* funding of the unique installation BIOCAI: UNLC08-1E-002, UNLC13-13-3503.
European Regional Development Funds (FEDER).

## Competing Interests

Carlos Fernandez-Lozano is an Academic Editor for PeerJ.

## Author Contributions

- Adrian Carballal conceived and designed the experiments, analyzed the data, authored or reviewed drafts of the article, and approved the final draft.
- J. Pablo Galego-Carro performed the experiments, analyzed the data, performed the computation work, prepared figures and/or tables, authored or reviewed drafts of the article, and approved the final draft.
- Nereida Rodriguez-Fernandez analyzed the data, prepared figures and/or tables, authored or reviewed drafts of the article, and approved the final draft.
- Carlos Fernandez-Lozano conceived and designed the experiments, analyzed the data, authored or reviewed drafts of the article, and approved the final draft.

## Data Availability

The handshake data, dictionary and code are available at figshare: Fernandez-Lozano, Carlos; Carballal, Adrian; Rodríguez-Fernández, Nereida; Galego Carro, Pablo (2022): WIFI-HANDSHAKE: Analysis of password patterns in WI-FI networks. figshare. Dataset. https://doi.org/10.6084/m9.figshare.19362971.v1.

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
