# Peer review of "Wi-Fi Handshake: analysis of password patterns in Wi-Fi networks"

_PeerJ Computer Science, doi:10.7717/peerj-cs.1185_

## Round 0.1 · original submission · Major Revisions

Please enhance the literature review with recent papers. Although Reviewer 1 has requested that you cite PeerJ Computer Science references, I do not expect you to include these citations. If you do not include them, this will not influence my decision.

·

Basic reporting

The paper would be improved by addressing the following concerns related to basic reporting:

(1) There is inconsistent use of "commas [,]" and "periods [.]" throughout the paper to separate numbers containing many digits. For example lines 338 and 339 include four digit numbers and use neither a comma or a period to separate the digits. In the text most frequently commas a used to separate larger numbers into 3 digit groups. However, in the tables the "words" column uses periods to separate the large numbers.

(2) The numbers include in every table are center aligned. This makes the figures included in the tables difficult to compare against one another because significant digits are not lined up on top of one another. For example, in Table the nine and eight digit words column do not line up with another such that digits with the same significance are in alignment. This issue needs to be addressed in all tables containing numbers with a different number of digits in a column. Also numerous spelling errors exist in the paper and should be corrected. The most notable is the term 'analizing' used in a section header in line 270.

(3) In the replication crisis error an anonymized version of the data provided in the paper, the scripts used for analysis, and the scripts used to create the figures for the paper need to be provided to both the reviewers and to the readership. This ensures completely transparent analysis and makes the authors paper significantly more impactful as other researchers can build off (and cite the paper).

(4) The reference list seems dated. Almost half of the references are more than 10 years old (the bulk of those coming from before 2010) and there are no references included from the 2020 forward. Understanding effective means of securing and exchanging passwords is a very current research topic. In addition, a number of related PeerJ papers are missing from the list a short selection is below:

Vorakulpipat, Chalee, Sasakorn Pichetjamroen, and Ekkachan Rattanalerdnusorn. "Usable comprehensive-factor authentication for a secure time attendance system." PeerJ Computer Science 7 (2021): e678.

Fremantle, Paul, and Philip Scott. "A survey of secure middleware for the Internet of Things." PeerJ Computer Science 3 (2017): e114.

Yu, Jinghua, Stefan Wagner, and Feng Luo. "Data-flow-based adaption of the System-Theoretic Process Analysis for Security (STPA-Sec)." PeerJ Computer Science 7 (2021): e362.

(5) Figures 5 and 6 present data as pie charts with numerous very narrow slices. It is difficult to understand which label is associated with which slice. In addition, the pie charts are interleaved with the bibliographic references making it difficult to scroll through the references as they are cross-referenced in the paper.

(6) Figure 3 contains numerous data types with effectively the same color palette used to label them . As a result it is difficult to read/understand the figure. I would suggest the authors pursue a different layout, perhaps an interactive one with tool tips.

Experimental design

The paper would be improved by addressing the following concerns related to experimental design:

(1) It is unclear what the research question that the authors are looking to address in the paper. The authors discuss the state of the art with respect to passwords and a wifi handshake but do not in the abstract of the introduction explicitly define a research question and hypothesize how their contribution / efforts will address it.

(2) There are not any results that are conveyed. As mentioned before a hypothesis is not postulated and there are no results with statistical significant reported. The main contribution of the authors seems to be a contextualized grammar that minimizes the size of the dictionary with respect to the most used dictionaries in an effort to improve and unify the decryption capacity of the combination of all of the dictionaries. However, it is unclear the actionable benefit that this offers.

(3) As stated earlier there is insufficient detail, methods and research artifacts (code, scripts) to replicate the analysis performed.

Validity of the findings

The paper would be improved by addressing the following concerns related to validity of the findings:

(1) The paper would benefit from a limitations section describing the bounds of the analysis and the different threats to validity which might prevent it from generalizing.

(2) As noted previously the underlying data has not been provided.

(3) It is unclear what the actionable findings from the work presented here are. For an individual who is going to use wifi in a public setting or a security officer tasked with making things more secure what is an actionable workflow that follows from this work.

Reviewer 2 ·

Basic reporting

This paper seeks to provide a snapshot of the security of Wi-Fi access points in the metropolitan area of A Coruna. The literature review is comprehensive, with sufficient background and context.
However, some language are not clear enough, and I would recommend further polishing the language.

In line 382, what is Infraestructure? Does author refer to Infrastructure?

Experimental design

Overall the xperiment design is clear and well aligned with the research question of this paper.

For table 5 and 6, I would recommending using commas instead of periods to report numbers. In addition, it would be more straightforward to use same unit, e.g. XX minutes, instead of X day Y hour Z minnutes.

Validity of the findings

The conclusion is clear. However, it would be better if authors can discuss more on the real world impact of the findings, as well as future works.

Reviewer 3 ·

Basic reporting

I have the following comments:
The authors should pay more attention to the Introduction section to substantiate and clarify the novelty of the proposed solution.
The authors should explain the security vulnerabilities associated with traditional passwords and the security advantages of adding salt to the password.
The English in the present manuscript is not of publication quality and requires improvement. It should be written in grammatically correct, logically connected sentences.


The authors should reorganize the manuscript as follows:
The handshakes collection should be included in the Materials and Methods section and
The pattern analysis and minimum dictionary development in the Result section.

Experimental design

The authors should explain in more detail the steps used in the experiment.

The authors should define if the target protocol were WAP, WAP2, or both.

The authors should include the research question(s).

Which is the real problem?
a) weak security passwords
b) WPA2 protocol
c) absence of adding salt in the access point

Validity of the findings

The authors corroborate previous studies where password security is weak, but what is the novelty of this work.
Authors need to explain the number of passwords hashed, the number of frames collected, and the time to hack passwords.

Additional comments

no comment

---

## Round 0.2 · accepted · Accept

I am pleased to inform you that your work has now been accepted for publication in PeerJ Computer Science.

Thank you for submitting your work to this journal.

·

Basic reporting

My concerns regarding the basic reporting have been sufficiently addressed. The paper is now suitable for publication.

Experimental design

My concerns regarding the experimental design have been sufficiently addressed. The paper is now suitable for publication.

Validity of the findings

My concerns regarding the validity of findings have been sufficiently addressed. The paper is now suitable for publication.

Additional comments

Overall, my concerns have been addressed. The paper is now suitable for publication.